**Data Availability Statement:** All data underlying the findings in this paper are included in the paper.

**Funding:** This research was supported by a grant awarded jointly by the US National Institute of

# Soccer clubs as avenues for gender transformative socialization of adolescent boys in Cape Town and Mthatha, South Africa: A qualitative study

Yandisa Msimelelo Sikweyiya[1,2]*, Natalie Leon[3,4], Mark N. Lurie[4,5], Mandla Majola[5], Christopher J. Colvin[4,5,6]

1 Gender and Health Research Unit, South African Medical Research Council, Pretoria, South Africa, 2 School of Public Health, Faculty of Health Sciences, University of the Witwatersrand, Johannesburg, South Africa, 3 Health Systems Research Unit, South African Medical Research Council, Tygerberg, South Africa, 4 Department of Epidemiology, Brown University School of Public Health, Providence, RI, United States of America, 5 Division of Social and Behavioural Sciences, School of Public Health and Family Medicine, University of Cape Town, Cape Town, South Africa, 6 Department of Public Health Sciences, University of Virginia, Charlottesville, VA, United States of America

* yandisa.sikweyiya@mrc.ac.za

## Abstract

In this paper, we explore the gender socialization of adolescent boys in soccer clubs, and ask whether there are opportunities for integrating gender transformative elements into that socialization. This qualitative study involved 11 in-depth interviews and informal conversations with male soccer coaches from Gugulethu township and Mthatha town in the Western Cape and Eastern Cape provinces of South Africa, respectively. Data were analyzed using a thematic analysis approach. We found that the coaches felt that the adolescent boys in their soccer clubs faced serious social and emotional challenges, with the boys' poor socio-economic backgrounds and fragmented family structures being major contributors to these challenges. Most coaches also gave themselves the responsibility to try to address some of the challenges faced by their club members. To do this, they employed specific strategies, including creating an alliance with parents and professionals. In the process, the coaches engaged the boys on topics around respect, sexual and reproductive health, and avoiding alcohol, drugs, and involvement in criminal gangs. Some coaches also played a social fathering role to club members as a way of helping them to think differently about their lives, redirect risky practices, and reduce the chance for poor health outcomes. These findings highlight the role of soccer clubs and coaches as potential avenues for health- and equity-promoting gender socialization of adolescent boys.

## Introduction

Globally, gender scholars have underscored the need to engage boys and men in gender-transformative programming (GTP) to challenge gender inequality [1, 2]. Gender-transformation uses mechanisms that target and alter harmful and unequal gender norms, roles, and power

Mental Health and the South African Medical Research Council (grant number R01 MH106600). The content of this paper is solely the responsibility of the author and does not necessarily represent the official views of the US National Institutes of Health or the South African Medical Research Council.

**Competing interests:** The authors have declared that no competing interests exist.

relations. It challenges the causes of gender-based health inequities whereby men enjoy a variety of privileges over women [3]. This work is vital for supporting boys and men to construct non-violent masculinities [4, 5], reduce risky behaviors, and improve health-seeking practices [6–8]. Research suggests that GTP with boys and men could potentially shift practices and reduce the acquisition and spread of HIV and other poor health outcomes [2, 8–10].

Some gender-transformative interventions implemented in South Africa have shown efficacy in strengthening boys' and men's linkage and retention in HIV care [see 6] and improving their sexual and reproductive health (SRH) outcomes [11]. Scholars have identified the adolescent phase as a critical time in which to intervene in the development of health, gendered and sexual behaviors [9, 12, 13]. As Blum, Mmari & Moreau [14] contend, adolescence is a time when boys learn most intensively how to construct their masculinities in line with cultural and social expectations of being a man in their settings. Scholars argue that such learning occurs through gender socialization, a process of learning culturally and socially ascribed behaviors, positions, and ideals deemed suitable for individuals, based on socially assigned gender [15]. Gender socialization occurs through inter-personal relationships and interactions with peers and parents, the setting and circumstances an individual lives in, and the social institutions in which they live such as family, schools, media, and churches [16–18].

In many communities in South Africa, studies have shown that boys are socialized to have dominance and control over girls, including in sexual and romantic relationships [19]. Moreover, boys are socialized to perceive that they are stronger than girls, undertake tasks that are more challenging, and have the freedom to experiment with heterosexual sex [19]. Boys are also given more social freedom and leisure time, while they are rarely expected to fulfil household chores [19, 20].

In this paper, gender socialization is understood as a process whereby a social agent (e.g., soccer coach) helps to socialize an individual (e.g., adolescent boy) into a local gender order (i.e. how gender is arranged in a particular place and time) [21], thereby helping to train them into a gendered world. Gender socialization is a process that is independent of any specific gender norms. Boys can be socialized into gender-equitable norms or gender-inequitable norms (or a combination of both). Gender socialization in soccer clubs thus refers to all the ways in which participating in soccer might socialize adolescent boys into a particular gender order, not only by explicitly promoting certain gender beliefs but also by showing boys how to relate to each other.

Some have argued that soccer clubs are ideal vehicles for gender transformative (GT) socialization among adolescent boys in South Africa [22, 23]. A growing literature suggests that important gender socialization work is indeed happening among adolescents in soccer clubs [22, 24]. However, not all that socialization is GT. Some of it may reinforce inequitable gender norms (e.g. men's superiority, control, and dominance over women) and dominant or destructive masculinities that emphasize power and force [25]. Gender work in soccer clubs could also promote masculinities that value sexual prowess and conquest in heterosexual relationships, practices that are associated with multiple and concurrent sexual partnerships, sexual risk-taking, unprotected sex, and sexual assault on intimate partners [12]. Indeed, as soccer is a tough, physical sport, it possesses the potential to reinforce some of the more damaging elements of local hegemonic masculinity. Yet, while strength and toughness are traits that are associated with local hegemonic masculine norms, it is not a straight or inevitable line from a more general characteristic (like toughness) to a specific underlying and toxic form of masculinity [26]. Furthermore, while soccer players are expected to demonstrate physical strength and toughness, there is also an awareness that for them to be successful in their soccer careers, they need to have good character, and self-control, be able to restrain themselves, and be disciplined [27]. As such, youth sports programmes, including soccer clubs, are known for teaching

discipline and etiquette to players including being on time for practice, respect, compassion, and caring [28].

We need to understand how gender socialization happens in soccer clubs if we want to take advantage of the potential opportunities soccer clubs offer as a potential pathway to transformative gender work. By keeping adolescent boys engaged, enabling them to exercise and be healthy, giving them a sense of achievement, and being part of a larger group, learning about team play, problem solving, practice for improvement, and providing them with a sense of hope for a better future, soccer clubs can profoundly shape the gendered norms and practices of adolescent boys [24].

The relationship between the players and the coach also contributes to gender socialization. Coaches can serve as positive or negative role models, or implement discipline in punitive or positive ways, as shown in Medich et al.'s study [22]. When these relationships and processes of disciplining and role-modeling are particularly intensive and evolving, scholars have sometimes framed them through the lens of 'social fathering'. Coaches can play a fathering role in the lives of adolescent boys [29]. The concept of social fathering applies whether a child has a biological father or other father figure at home, and the role is often played by men in local social institutions such as the church, school, or soccer club. In this article, we use the term as a way of talking about the ways male coaches who are not biological fathers can play both a fathering and gendering role. The idea of social fathering is not unique to soccer clubs; rather, it is part of the general way of thinking about more distributed forms of fatherhood in South Africa [30] and our understanding of soccer coaches as social fathers is limited. Research on social fathering in South Africa has mainly been done with adult men in community-based studies [30]. There is little work referring to the coaches themselves and their experiences in the social fathering role.

This paper explores how gender socialization works in soccer clubs, and whether there are opportunities for integrating GT elements into that socialization. The dynamics between coaches and boys are unpacked from the coaches' perspective, to learn more about how the coaches are—or could be—playing a mentoring role in supporting the gender socialization of adolescent boys. Moreover, we explore the extent to which this socialization involves a social fathering role (as opposed to, or in addition to other mechanisms for gender socialization).

## Context of the study

This study was part of a series of case studies conducted under a larger project called iALARM (Using Information to Align Services and Link and Retain Men in the HIV Cascade), a 5-year, NIH-funded research project on men and HIV. As described by Colvin and colleagues [31], iALARM's central intervention was the creation of a 'Linkage and Retention in Care Task Team'. The Task Team was comprised of various stakeholders including representatives of the local clinic, staff and managers from other health facilities, officials from local government sub-district offices, and representatives of several local non-governmental organizations (NGOs). The NGOs involved in the Task Team included ones focusing on child and adolescent wellness, youth sports (including soccer clubs), parenting and community health, food security, and disability [31]. This study is one of several sub-studies within the iALARM project that examined community- and health system-based strategies for mobilizing men and boys around HIV and gender equity.

## Material and methods

### Study design

This was an exploratory qualitative study that gathered data through in-depth interviews (IDIs) and informal conversations with soccer coaches. The study design was guided by our

interest in developing an in-depth appreciation, from the perspectives of adult male soccer coaches, of the following issues: their coaching background; the background of the adolescent boys in their clubs; the main issues that come up in coaching; their lived experiences balancing the task of training soccer with the work of teaching life lessons within clubs; addressing the social well-being of boys; dealing with boys' experiences of challenging and disadvantaged home backgrounds, school and community environments; interpersonal and emotional issues boys encountered; boys' dating and sexual relationships; and the strategies coaches implemented in socializing adolescent boys in their clubs. The interview guide for the IDIs was framed around these questions (see S1 File).

## Setting

In the Western Cape Province, data collection was conducted with soccer coaches who resided in the township of Gugulethu—a peri-urban township with approximately 100,000 people [32]. South Africa's high unemployment rate of 29.1% [33], and very high youth unemployment are long-standing problems. Gugulethu is characterized by high levels of poverty and high rates of interpersonal violence including homicide, gangsterism, and substance abuse. In the Eastern Cape Province, data collection was conducted with coaches who lived in townships and villages in and around Mthatha—a small town in the OR Tambo District. The province has the highest unemployment rate in South Africa at 37.4% [33], and Mthatha is characterized by limited economic resources, a high crime rate, harmful use of alcohol, and the use of illicit drugs among the youth [34–36].

## Sampling and participants

Eleven Black African male coaches were purposively selected to participate in IDIs. First, during the iALARM Task Team meetings, we approached soccer coaches who were present, informed them of our research and its purpose, and invited them to participate in one-on-one interviews. All coaches who were approached agreed to participate and were interviewed, and thereafter offered to share names and contact details of other coaches who were part of their social networks in Gugulethu. We then called the suggested coaches and those whose soccer clubs were training adolescent boys were invited to participate in the study and were also interviewed. In total, five coaches from Gugulethu and six coaches from Mthatha were interviewed. The coaches were between the ages of 30–50 years, all had completed grade 12, four had a tertiary qualification, and six had formal soccer coaching certificates of different levels. The coaches' experience in training adolescent boys ranged between three and 23 years. Coaches reportedly spent approximately five days a week in this role during the soccer season.

## Data collection procedures

The data were collected through IDIs and informal conversations with the soccer coaches. IDIs were conducted either in-person in a community hall or telephonically in isiXhosa — the dominant spoken language in Gugulethu and Mthatha.

We stopped conducting the interviews at interview 11 for two reasons. First, there were a limited number of soccer coaches who trained adolescent boys in both Gugulethu and Mthatha. Second, when we reached interviews 9 and 10, we felt that we had reached data saturation. The digitally recorded interviews were conducted by the first author who is an experienced qualitative researcher, male, multi-lingual, and a first language isiXhosa speaker who originates from Mthatha. The recordings were transcribed verbatim and simultaneously translated into English by the first author in preparation for analysis.

## Data analysis

Data were analyzed inductively using a thematic analysis approach [37], guided by the research question and interview categories of interest. Yet, there were deductive elements to the analysis [38]. The first (YMS) and last author (CJC) were both involved in various stages of data analysis. We first read and re-read the transcripts individually to familiarize ourselves with the content of the transcripts. Next, we established general codes which somewhat resembled the questions in the interview guide. Next, the first author examined the data and identified several further 'open' codes, which provided information beyond the initial interview questions, which he discussed with the last author. Through an iterative process, open codes were discussed and those considered to be alike were grouped under clearly distinct themes. When there were discrepancies in the codes, we resolved them by verifying the codes against the data. Afterward, we assessed the patterns and links between the themes and interpreted what we saw emerging.

## Ethical considerations

This study was approved by the University of Cape Town's Faculty of Health Science's Human Research Ethics Committee (#802/ 2014 and 655/2016). Before conducting the IDIs, the first author reviewed the participant information and consent form with all participants. The discussion included the rights of participants and the risks and benefits of participating in the study. Verbal informed consent was obtained from all participants before they were interviewed. Pseudonyms were used to protect the identity of participants. No financial incentive was given to participants for study participation, and this was communicated to the participants before conducting the interviews.

## Results

The results have been structured into two main sections. The first section describes how coaches perceived the problems adolescent boys were facing and the underlying reasons for these problems. The challenges included social and behavioral problems, emotional stresses, and class tensions within their communities. The second section explores how coaches, by playing a social fathering role, contributed to the gender socialization of the boys in their soccer clubs. Coaching strategies included forming supportive relationships with the soccer players, as well as fostering relationships with parents and professionals to address gendered-sensitive topics relating to respect, sexual and reproductive health (SRH), alcohol and drug abuse, and involvement in crime.

## Coaches' perceptions of adolescent boys' problems

The coaches identified three problem areas and underlying dynamics associated with these problems—social and behavioral problems linked to substance abuse and gangsterism; emotional problems related to family dynamics and home stressors; and class differences related to socio-economic background.

**Social and behavioral problems boys were facing.** Almost all the coaches said some boys in their clubs were experiencing significant levels of social and behavioral problems, including the use of illicit drugs, harmful use of alcohol, and involvement in gangsterism:

> I have two boys in the team who were involved in drugs before the age of 15, but I was able to help them because I met with their parents. (Bafo, Gugulethu, Township)

Some coaches spoke about their efforts to discourage adolescent boys from being involved with criminal gangs, in their townships:

> I took [gangster] boys from the township and mixed them with boys in my club and after all that effort other coaches were asking how I did that. And after that [intervention], the issue of gangsterism has just disappeared . . . I aimed to fight the use of drugs in the community by targeting those who are known drug users, to form their own team, and if they win, we would buy [soft] drinks and bread because giving them money is not ideal. (Tony, Gugulethu, Township).

According to these coaches, the boys' 'difficult backgrounds' and 'challenging lives' were linked to the social problems they were facing:

> You know, [years back] you would never struggle to find 12-year-olds and 14-year-olds to join your club. But now [problems] start from 13 years; they start having girlfriends and be [ing] absent from school. You will find that these things used to happen among 17 or 18-year-olds, but now they start at a younger age, and from ages 17–19, they are already ruined. (Masixole, Mthatha, Township)

Most coaches identified traditional male circumcision (TMC) as one of the key mechanisms for socializing boys. However, several of the coaches believed that the problematic behaviors of adolescent boys could also be related to changes in the practice of this rite of passage to manhood. They felt that TMC presented a threat to healthy adolescent development and behavior. They argued that it no longer performs the positive, socializing role that it did in the past.

For the coaches, the changes they were observing in TMC processes, compared to earlier times, were producing poor social and health outcomes for newly circumcised men (*amakrwala*). Specifically, they highlighted the introduction of new practices (e.g., use of illicit drugs and alcohol, and the 'Numbers gang' language) into the custom:

> You raise a child well, make him play soccer. He goes there [initiation school] and is told to smoke, "A man smokes". They drink [alcohol] there, so the child comes back in a worse condition. (Bunono, Mthatha, Township)

Having perceived that TMC was no longer performing its intended role of socializing boys into becoming responsible and respectful men, several coaches said this motivated them to actively intervene in the lives of the boys to positively shape their behaviors and masculinities.

**Emotional challenges faced by boys.** The coaches further identified underlying emotional problems experienced by the adolescent boys, which they related to the absence of positive father figures and other stressors in the home environment:

> His dad separated from his mother when they were young so all he needed was a father figure to guide him. I said to his mother, "A child must be close to other children and because he was wearing old torn clothes, if you have a problem send money to me then I will accompany him to buy clothes". All he was lacking was a father figure, . . . now he is doing Grade 12. He is updating me every time about his progress. (Xola, Mthatha, Township)

Similarly, many other coaches perceived boys as needing a father figure in their lives. On recognition of this need, they felt it was their responsibility to step into the social fathering

role. While not all the coaches said they played a social fathering role, most gave accounts where they actively played this role:

> The [biological] father loved that boy; he was supportive although he was abusing alcohol. The father got sick and died. After his death, things started to get worse for the 16–17-year-old boy. The boy would go to school and come back very late. His mother reported the boy to me that he once came back from school drunk. Then I tried to play a father's role to the boy. (Masixole, Mthatha, Township)

Most of the coaches told us that some of their boys came from unstable, conflictual, and punitive home environments and were facing emotional problems due to hardships associated with these family situations. For example, witnessing fights between their parents, family dissolution, being physically beaten or deprived of food at home, and other hardships were common at home:

> The issue that these boys have come to us [coaches] for, is the situation in their homes. Sometimes we would go to their parents and ask what is happening because we notice that he [boy] is no longer the person he used to be in the field, and we don't want to continue like nothing is happening . . . We tell parents that we are not trying to teach them 'how to treat your child but if you are punishing him by not giving him the food, he will go and seek that outside in the streets'. (Thobani, Mthatha, Rural village).

Masixole narrated a story of a boy in his club who was abused by his father and deprived of food as a punishment for not fixing cars:

> There was a boy in my club, his father was fixing cars . . . so when returning from school the boy was expected to fix the cars before he could go to the gym. He had no soccer boots and would steal other children's belts so that he could sell them to get some cash, so when you asked him why he steals, he would say "it's not easy at all, his father is not giving him food if he didn't repair any car". The boy was 19, ended up being a criminal, who was shot and killed by police officers. (Masixole, Mthatha, Township)

Masixole said he had tried to intervene with the boy to stop his criminal behavior, but the boy explained that his behavior was caused by his difficult experiences at home.

**Class spectrum in townships.** The third area of underlying problems was class differences due to different socio-economic backgrounds, with some boys coming from disadvantaged backgrounds, while some were coming from relatively well-off families:

> I need to have an eye on other things apart from football. I created an emergency fund so that if we are going to a match, I put an R100 [∼US$ 6] aside for them to eat after the game. . . the area I reside in has a mix, but if you consider the resource-poor townships, you see all those who come from that side you can tell even by their clothes that 'there's a shortage of something'. (Xola, Mthatha, Township)

Most coaches perceived boys coming from well-off and stable families as being less vulnerable to most of the problems described earlier:

> And it's a vice versa, he [another coach] was close to rich players and I was close to poor players, so that was the reason why his team was not consistent because he was not able to protect these [poor] boys. (Masixole, Mthatha, Township)

In recognition of these class-based differences among the boys, most coaches talked about putting more effort into making the poorest boys feel welcome and comfortable in the clubs, availing themselves more emotionally to them while also providing them with material support, whenever there was a need:

> Anyway, I saw it [emergency fund] is needed by the time I was paying for this boy at Vuku-zithele Senior Secondary School because he was going to be expelled there, they wanted R500 [∼US$ 30] so I said they must not expel him from school, I took this money and paid. . . (Xola, Mthatha, Township)

On the other hand, the coaches perceived the boys who were well-off as not needing much support beyond being trained for soccer, even though they tried to form a good player-coach relationship with these boys. For these coaches, it was essential to maintain a good balance in how they treated the boys from the different ends of the class spectrum, for the club to function well.

## Social fathering and gender socialization

The social and behavioral problems experienced by boys were not straightforward and dealing with them was difficult for the coaches. Yet, most coaches mentioned that, through their clubs, they ended up doing a great deal of work to help socialize adolescent boys and to positively shape their practices and support them in constructing alternative masculinities.

**Coaches as social fathers.**   When the coaches looked at the problems and challenges experienced by some boys in their clubs, one of the most common reactions was being social fathers to them:

> So that makes me think the attention we [coaches] are giving them [boys], they do not get at home, so we are no longer just coaches we are also the fathers, like for instance when a boy says they do not have food at home so you must assist where you can . . ., so it's not just coaching it's nurturing and fathering the boys. (Thobani, Mthatha, Rural village)

For the coaches, playing this role was a significant intervention to help the boys change their journeys and think differently about their lives; to encourage them to change their behavior and practices, and for the coaches to mitigate the risk of poor health outcomes in the lives of these boys:

> I saw an opportunity, because they had chosen to play soccer instead of smoking weed [marijuana], so I thought if I don't do anything about this, they will end up going to do things that aren't right. I ended up telling them that "as of today you will be playing for my club . . . and you must know that playing football is not the only important thing, the other things that are important are respecting other people, education, working hard and taking life seriously". (Thobani, Mthatha, Rural village)

In playing the social fathering role, the coaches served as role models to the boys by behaving appropriately in their presence, including not drinking alcohol or smoking marijuana:

> I am not a person who behaves badly in the area. . . I would not walk in the street carrying alcohol. So even if I do those things, I do it out of my boys' sight. When I talk to them, I ask if they have ever seen me walking around carrying beer and they say no. So, I say "when

your mother sees you carrying a beer, where will you say you learned that from?" (Ayanda, Gugulethu, Township)

Other coaches more deliberately engaged in gender socializing work with the boys as they spoke with them about 'how a man is expected to behave', including the need to respect women:

They [boys] don't know what it means to be a man and how a man should behave, which is why we teach them that they must be respectful to the next person and women and this thing of calling women names must end. They must also respect their parents, they must always be clean, we are trying to instill the values, they must always remember what he was taught at the club. And we don't want anyone to drink or smoke in this team. (Thobani, Mthatha, Rural village)

The coaches mostly had a hierarchical authority role in the clubs. However, their relationships with the boys were also more dynamic and less structured than expected and often did not fit the assumed rigid and neatly arranged relationship of the soccer coach-player. While some boys, especially those from financially and socially stable home environments, tended to forge simple relationships with the coaches which involved less nurturing by the coaches, boys from poor families or dysfunctional homes appeared to develop a more emotionally involved and nurturing relationship that went beyond coaching:

We [coaches] must intervene in such cases and play a father figure and even on WhatsApp they would take my profile pictures and post them on their walls and say 'this is my father'. They would say 'my life changed ever since I met him, he is a coach, he is a motivator and a spiritual father', so you would see that indeed he missed that father figure in his life. . . (Ongama, Mthatha, Township)

The relationships with boys from financially impoverished and unstable family environments almost always extended to the 'private space' of home and deeper involvement (e.g., in disciplining methods and schooling) by the coaches. Yet, these relationships were largely not formally agreed to, causing coaches to be uncertain how involved they should be in the boys' lives, and sometimes their involvement was not welcomed by parents.

**Social fathering and the family at home.** The social fathering role did not replace the boys' relationships with their family but existed in parallel to those family relationships. Coaches described several ways in which they tried to work in synergy with the parents to shape the boys' behaviors, masculinities, and lives:

Our academy has parent leaders called the 12th player with a WhatsApp group where parents report and discuss anything concerning the players. (Ongama, Mthatha, Township)

The coaches shared that while not all parents were eager to partner with them, some parents were willing to work with them in addressing their children's challenges:

Yes, because now there are a lot of diseases and they [boys] don't know about that. I don't want parents to say I am teaching their kids about dating, so we need to talk to these kids together. (Tony, Gugulethu, Township)

While most coaches said parental involvement in clubs was rare, a few coaches mentioned that some parents were involved, had the interest to know what happens in the

clubs, and were keen to work in tandem with the coaches in socializing the boys and shaping their behavior.

> There was a boy in my club, he became involved in gangsterism at school. So, to punish him, his mother said he must stop playing football. I said to the mother 'please give me a chance with this boy, I'll put him in a programme then if he is not responding to the programme, you can take him . . . then I sat down with this boy and told him 'if you want to grow and be independent just like your parents you need to prioritize school', and I realized that counseling should be involved in this . . ., so we continued with this, and you won't believe he is doing Grade 12 now. (Xola, Gugulethu, Township)

The coaches said they observed behavior changes in some boys, including constructing health-enhancing masculinities, when they worked closely with their parents. However, some coaches talked about being perceived by parents as encroaching into spaces that are not only private but also out of bounds for them. Indeed, the coaches described instances where their attempts to involve themselves in socializing the boys, beyond 'just training them soccer', brought tensions in some families:

> The punishment back then was that when the team is going to play, the boy being punished will take the soccer kit home and wash it. The boy's mother was not happy with this, so I ended up stopping with the punishment. (Masixole, Mthatha, Township)

Consequently, while these coaches felt compelled to involve themselves in socializing the boys, they recognized they were constrained in doing so by the tensions their involvement brought, and the subtle feedback from some parents who felt the coaches were 'going too far and involving themselves in matters that do not concern them'.

> His [player's] mother called us [coaches] and she said she feels like this boy's studies are being disturbed by him playing soccer. . . We never did much to convince her that the club could find him a teacher who can provide him with academic support, we never did because we saw from her response that she was not open to working with us. (Ongama, Mthatha, Township)

Social fathering was a common, but complicated role for coaches to take on, one that required constant and sometimes complex negotiation with parents about how best to support the boys in their club.

**Professional support at the limits of social fathering.** Coaches also described confronting critical limits to their (or the boys' families) ability to fully support their club members. For example, some coaches felt like they didn't have enough information to enable them to play this supportive fathering role effectively:

> I promise you from now on we will try to be open to them [boys]. . . it's just that we are not qualified for these things [SRH issues, sexuality, and dating]. . . sometimes there are things that you are afraid to talk about. (Bafo, Gugulethu, Township)

Coaches spent a considerable amount of time with the adolescent boys, especially during the soccer season, and they were able to get to know some of the personal and social challenges faced by individual boys. This deeper level of interpersonal engagement is what appears to have motivated the coaches to take on a social fathering role. Yet most coaches felt that they were unequipped to manage this role:

> It was a team of under-17s that was playing very well and got promoted, but we would only have four of them attending training . . . it was *amapara* [illicit drug users] and they would come to practice smelling of dagga and intoxicated with drugs. So, if there is no psychologist to help you, then you won't know what to do, and you don't get any cooperation from the parents because they don't have a passion for their kids to play soccer, you see? (Viwe, Mthatha, Rural village)

A few coaches talked about using professionally trained people to engage the boys in matters that they felt were beyond their capabilities. These coaches reportedly had a standing agreement with some professionals or institutions to come to their clubs at certain times and address issues affecting adolescent boys:

> We have a medical doctor, who assists with the medical side of things. We also have a psychologist, who comes once a month and have sessions with boys who need it. The psychologist works with Nomsa, who is a life coach. She deals with how they [boys] should conduct themselves in life and motivates them. (Ongama, Mthatha, Township)

Another coach asserted that it would not be difficult to engage boys from his club on SRH issues affecting them, yet he thought this was beyond his core function as a coach and that he would prefer to be assisted by a health professional:

> My engagement with the boys is mainly about soccer–and the boys only perceive me as a soccer coach. I feel that as part of the strategy to engage the boys about health and sexuality issues, someone who understands health challenges affecting boys would need to come and talk with the boys. (Thabo, Gugulethu, Township)

There could be various reasons the coaches preferred to work with professionals in engaging boys in these issues. First, they may have perceived it as a viable strategy to connect the boys to significant resources, but also draw support for themselves as they had limited capacity to address all the challenges on their own. Alternatively, the coaches may have felt overwhelmed and/or underqualified to address these issues, hence reaching out to those with expertise to support them.

## Discussion

In this study, we explored how gender socializing works in soccer clubs, specifically assessing the potential of soccer clubs as approaches for GT socialization of adolescent boys. We also explored the degree to which such socialization involves a social fathering role played by coaches, and how that may have opportunities for having GT elements in that socialization.

Broadly, the findings of this study provide useful insights into how gender work with adolescent boys can be done within soccer clubs. Our findings suggest that soccer clubs are potentially valuable spaces for socializing gender transformation, and social fathering is an important component of that if it is going to be successful. We have demonstrated that soccer clubs and coaches can be significant socializing spaces and agents, respectively, for adolescent boys in Black South African communities. Moreover, we have shown that coaches were eager to and do play the social fathering role, yet playing this role is complicated. They faced challenges with playing a social fathering role related to balancing the role of coach and social father in the relationship with the boys, limited knowledge on how to address problems, and difficult relationships with parents. They outlined various strategies for addressing the multifaceted social, emotional, and health problems they perceived among boys, especially those

from poorer and dysfunctional families. Key strategies were the alliances the coaches forged with parents and professionals. Our findings show that social fathering is complex and that it needs support, resources, training, and skills. Moreover, it requires the engagement of others as the coaches cannot do it all by themselves.

Through playing the social fathering role, the coaches intervened when the boys were engaging in destructive or risky behaviors (e.g. alcohol and drug use) and masculinities. These findings corroborate Medich et al.'s [22] findings that coaches in Cape Town townships played the roles of a father to boys in their clubs by giving them advice and guidance on how to address the challenges they were facing, including advice on avoiding involvement in gangsterism. Our findings show how the coaches' interventions happened through educating and supporting (emotionally and socially) the boys to alter their journeys. They challenged the boys to critically reflect on their risky behaviors and practices, approach life in a more progressive way (e.g., work hard at school, and be respectful towards women), think differently about their [difficult] life circumstances, and set new progressive life goals. Some coaches more directly engaged in gender work with the boys, challenging destructive masculinities and gender norms. This gender socializing has the potential to mitigate the risk for poor health outcomes for boys. With the right support and resources, the social fathering role played by these coaches has the potential for GT socializing as well, to support the development of more equity-based gender norms. To achieve this GT socializing, the coaches would need to be more supported by the parents of the boys and by professionals and broader community support organizations.

Soccer is popular in most Black African communities in South Africa [39] and soccer clubs are considered vital for socializing adolescent boys. In our study, a key factor that motivated the coaches to get involved in soccer clubs and play the social fathering role was the widespread perception that some of the other ways of socializing adolescent boys (e.g. TMC) are failing [40, 41]. Historical literature from South Africa suggests that in the past TMC succeeded in its intended role of socializing young boys into local norms of respectful, responsible men because of certain elements it possessed [42, 43]. Hellman [44] has noted that, long time ago, TMC in Black rural communities was more of a group process that enabled a sense of an age cohort — with initiates positively influencing each other. Since the past few decades, however, TMC has become more of an individual process. Wilson and Mafeje [45] observed in Langa township, Cape Town, that nowadays the emphasis is put on the clinical operation (circumcision), with very little focus on the educational aspect of the custom and that the period of seclusion for initiates is shorter [see also 46]. This limits the time for the initiates to be educated on moral values such as self-respect and respect for others, and social responsibility [45].

Owing to their perception that TMC was no longer performing its intended socializing role, the coaches took on the responsibility of socializing adolescent boys, with many becoming social fathers to boys in their clubs. However, despite their eagerness to play this role, doing so came with several complex challenges. First, some coaches felt they lacked the capacity to play this role, citing a knowledge gap on the approaches for addressing the social and health problems adolescent boys face. Second, for some coaches, playing this role resulted in them having conflicts with the families of some boys, as the families felt that the coaches were overstepping the boundary.

However, our findings suggest that some boys reportedly appreciated the social fathering aspect of their relationships with the coaches, with some publicly (through social media platforms) acknowledging the coaches as father figures to them. This finding reflects that reported in a study among young Black men in informal settlements in Durban who said they cherished the 'guidance on growing up' that they received from social fathers in their communities [47]. We argue, however, that it is unreasonable to expect coaches to play all these fathering roles to boys in their clubs. While all the coaches were prepared to play this role, most felt they lacked

the capacity to do so, and that they needed more skills and support. Because of their inadequate skills, knowledge, and training, they reached out to parents and professionals to try to fill these gaps.

We found that a typical model for a coach, in the study settings, is that they become social fathers who provide guidance, discipline, and emotional and social support to adolescent boys in the clubs. While this makes coaches important father figures to some boys, a more salient finding is that the coaches catalyze new relationships for the boys. They become brokers for new alliances and relationships between the boys and their parents and professionals, which, in turn, become useful sources of support for the boys to address their social and emotional challenges.

There is a growing body of literature in South Africa focusing on soccer clubs as spaces for gender work [22, 24]. Medich and colleagues [22] assert that 'soccer is a facilitator for friendship bonding, for discourses of masculine agendas, and for larger discussion on gender dynamics' (p.196). Our findings add to this body of knowledge by showing how soccer coaches engaged in the process of gender socializing adolescent boys in resource-limited communities. In our study, the coaches emphasized the significance of self-respect, respect for women and elders, discipline, and positive contribution to society. Through such gender work, the coaches were intending to shape and reinforce a positive soccer masculinity [48] among the boys, one that will make both their coaching careers and boys' soccer careers successful. However, in engaging in this gender work, it is unclear whether the coaches had made a conscious effort to shape a new gender-equitable male youth. This warrants further exploration through in-depth research on the coaches' role and motivations for engaging adolescent boys in sports.

Our findings support South African research that has shown that work on gender is already happening in soccer clubs in Black communities [22]. We, therefore, argue that soccer clubs present an opportunity for researchers and programmers to use them as vehicles for behavior change [49] by leveraging the gender work already occurring in these clubs, and ensuring that such work is informed by progressive theories of gender, power, and masculinity. Our findings also suggest that soccer clubs offer a useful entry point to draw adolescent boys away from social vices, yet we contend that to succeed in re-socializing adolescent boys into health-promoting norms of gender, structural changes that incorporate a focus beyond sports are needed. In South Africa, there is an urgent need for structural interventions aimed at improving the livelihoods of working-class people in resource-poor communities. Most Black South African communities are characterized by high levels of poverty, unemployment, and dysfunctional families [50]. Thus, our finding on absent or uninvolved fathering in the families of some boys in soccer clubs should be understood in the context of economic disenfranchisement of Black people which has its roots in colonial and apartheid systems [51], and the continued socio-economic, socio-political, and structural forces in democratic South Africa that continue to adversely affect and shape the lives of adolescent boys in poor Black communities.

## Implications for programming and research

Our findings highlight the practice of soccer clubs and coaches as avenues for gender socialization of adolescent boys that is aimed at promoting their health and social well-being. It provides an opportunity to reflect on how these same gender socializing mechanisms could be supported as ideal opportunities for GT socializing. To achieve this, soccer clubs and coaches may benefit from working closely with and drawing support from NGOs, and sport and youth development initiatives that provide GT programmes and parenting support interventions.

Research is needed to explore the actual gender politics of the coaches and in their interactions with adolescent boys. Gender work is already happening within soccer clubs, yet whether

that work reinforces conservative gender norms and masculinities warrants further research. The notion that soccer coaches can play a social fathering role as part of GT interventions is a key assumption that requires further examination of the kind offered in this paper. To strengthen this evidence base, future studies should interview adolescent boys in soccer clubs as that would add an important dimension to our understanding of both the gender socialization happening within soccer clubs and how this can be shaped to be gender transformative. Last, there is a need to understand the longer-term impact and sustainability of soccer clubs as potential GT vehicles, and what specific aspects of gender socializing might be helpful or harmful for the health of adolescents.

## Strengths and limitations

The scope of our study is limited on the question relating to the ultimate boy-product the coaches wanted to see when they engaged in the social fathering role with the boys in their clubs. As such, our data do not allow us to adequately address this question in this paper. Future studies exploring the social fathering role played by soccer coaches and how it may work to socialize boys into a gender order in a particular setting would do well to explore this question. Exploring this question is critical as it lies at the heart of transformative gender work with adolescent boys.

The purposive selection of coaches may have been more biased towards coaches that were more inclined towards supporting socializing of their soccer team members, and the transferability of the findings may be limited to similar settings. The strength of the study is that it included coaches from urban and rural settings which gives it broader transferability. Coaches provided a nuanced view of their experiences, including being open about their challenges, and this together with the communalities across the two settings, increases our sense of the credibility of the findings. We hope these findings would be used to generate hypotheses for future research with adolescent boys in soccer clubs in South Africa and elsewhere.

## Reflexivity

The authors are researchers who are situated within both academia and practice. Authors (YMS and MM) were born and had lived in the two study settings, as adolescent boys, and amateur soccer players and were able to provide an insider perspective on this topic. Their prior experiences in these settings may potentially bias their views on the analysis of the data. Other authors (CJC, NL and MNL) are White academics who have spent many years in the Western Cape province where they conducted extensive health systems research, community development work, health promotion work, and GT interventions with boys and men in Black poor communities in the Western Cape province. As such, while they have extensive knowledge on this topic and are highly skilled in qualitative data analysis and interpretation, their personal experiences, identities, and social positions may have influenced how they interpreted the data. Notwithstanding, the discussion and refining of interpretations across the author team ensured that the findings and interpretations reported here were grounded in the data.

## Conclusions

In Black African communities in South Africa, families are viewed as vital spaces where boys get gender socialization. However, in resource-poor communities, there is a perception that the conventional strategies of socializing adolescent boys have been compromised and are failing in this role. Gender socialization of boys can be done either conservatively or progressively, and soccer clubs represent an opportunity for the latter approach. We have shown here that gender work that is health promoting and produces a soccer masculinity among adolescent

boys is already happening in clubs, and that this could lay the foundations for GT work. However, coaches need to be capacitated and supported to ensure that gender work within their clubs is progressive.

## Supporting information

**S1 File. Interview guide.**
(DOCX)

## Acknowledgments

The authors would like to thank the participants who shared their stories, insights, and experiences with us. We thank Lenadine Koza for editing the final version of this paper.

## Author Contributions

**Conceptualization:** Yandisa Msimelelo Sikweyiya, Christopher J. Colvin.

**Data curation:** Yandisa Msimelelo Sikweyiya, Christopher J. Colvin.

**Formal analysis:** Yandisa Msimelelo Sikweyiya, Natalie Leon, Mark N. Lurie, Christopher J. Colvin.

**Funding acquisition:** Christopher J. Colvin.

**Investigation:** Yandisa Msimelelo Sikweyiya, Christopher J. Colvin.

**Methodology:** Yandisa Msimelelo Sikweyiya, Christopher J. Colvin.

**Project administration:** Yandisa Msimelelo Sikweyiya.

**Resources:** Christopher J. Colvin.

**Supervision:** Christopher J. Colvin.

**Validation:** Natalie Leon, Mark N. Lurie, Christopher J. Colvin.

**Visualization:** Christopher J. Colvin.

**Writing – original draft:** Yandisa Msimelelo Sikweyiya, Natalie Leon, Mark N. Lurie, Mandla Majola, Christopher J. Colvin.

**Writing – review & editing:** Yandisa Msimelelo Sikweyiya, Natalie Leon, Mark N. Lurie, Mandla Majola, Christopher J. Colvin.

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
