## [Decision Letter · Decision Letter 0]

10 Dec 2021

PONE-D-21-26547The potential of soccer clubs as avenues for gender transformative socialization of adolescent boys in Cape Town and Mthatha, South Africa: findings from a qualitative studyPLOS ONE

Dear Dr. Sikweyiya,

Thank you for submitting your manuscript to PLOS ONE. After careful consideration, we feel that it has merit but does not fully meet PLOS ONE’s publication criteria as it currently stands. Therefore, we invite you to submit a revised version of the manuscript that addresses the points raised during the review process.

1. Specifically, authors need to explain why they did not explore these issues among boys themselves.

2. Given soccer is a tough, physical sport, does it not reinforce hegemonic masculinity?

3. Authors need to present their understanding of concepts (e.g. gender socialization.

4. Finally, consider the WHO definition of a gender transformative approach i.e. one ‘that addresses the causes of gender-based health inequities through approaches that challenge and redress harmful and unequal gender norms, roles and power relations that privilege men over women’

We look forward to receiving your revised manuscript.

Kind regards,

Webster Mavhu

Academic Editor

PLOS ONE

Journal Requirements:

3. In your Data Availability statement, you have not specified where the minimal data set underlying the results described in your manuscript can be found. PLOS defines a study's minimal data set as the underlying data used to reach the conclusions drawn in the manuscript and any additional data required to replicate the reported study findings in their entirety. All PLOS journals require that the 

Reviewers' comments:

Reviewer's Responses to Questions

**Comments to the Author**

1. Is the manuscript technically sound, and do the data support the conclusions?

Reviewer #1: Partly

Reviewer #2: Yes

Reviewer #3: Partly

2. Has the statistical analysis been performed appropriately and rigorously? 

Reviewer #1: N/A

Reviewer #2: N/A

Reviewer #3: N/A

3. Have the authors made all data underlying the findings in their manuscript fully available?

Reviewer #1: Yes

Reviewer #2: Yes

Reviewer #3: Yes

4. Is the manuscript presented in an intelligible fashion and written in standard English?

Reviewer #1: Yes

Reviewer #2: Yes

Reviewer #3: Yes

5. Review Comments to the Author

Reviewer #1: Thank you for the opportunity to review this manuscript. The manuscript describes findings from a qualitative study which aimed to explore how gender socialization of adolescent boys works in soccer clubs, and whether there are opportunities for having gender transformative elements in that socialization. I think this article has the potential to make an important contribution to the literature, but it would benefit from further revision. Please see my comments below:

Major comments:

- The authors state several times that this study aims to explore how soccer clubs shape the gender socialization of adolescent boys. The authors then go on to claim that they achieved this aim. However, I’m not fully convinced. I think the manuscript would greatly improve if the authors first define what they mean by gender socialization and then provide more evidence to back up their claim that their findings demonstrate how soccer clubs shape gender socialization of adolescent boys. A lot of the examples in the results section that are described as socialization do not appear to specifically challenge hegemonic masculine norms or promote gender equitable norms. The findings focus much more on how the coaches address challenges related to the boys’ social and economic situation, which in my mind, is not necessarily linked to gender norms.

- I appreciate that the authors call for future research to understand these research questions from the boys’ perspective, but I think the fact that the authors did not interview boys for this project should be listed as a limitation. Having the boys’ perspective on their own experience of how the soccer club may/may not shape their gender socialization is critical to our understanding of this process. I think a rationale for why the authors only focused on the soccer coaches is also necessary.

- In the discussion section (page 24, lines 624-629), the authors state that “our findings show how the coaches’ interventions happened through role modelling, mentoring, educating and supporting (emotionally and socially) the boys to alter their journeys; they challenged boys to critically reflect on risky behaviors and practices, approach life in a more progressive way (e.g. work hard at school and be respectful towards women)…” After reading the results section, I was surprised by this claim because I did not see much of this reflected in the results. More evidence needs to be provided about how the coaches served as role models (and what behaviors or attitudes they role modeled) and how they taught boys to be respectful towards women. I did not see evidence of this in the results section.

Minor comments:

Abstract:

- The authors mention the iALARM Task Team in the abstract, but there is not explanation of what this is. We are only introduced to this term at the end of the introduction. A brief description of what this is should be included in the abstract to help orient the readers.

Intro:

- The authors use the term “gender” in several ways throughout the introduction that needs more attention to make sure the correct meaning is being conveyed. For example, on page 3, line 76, the authors use the term “To transform gender.” I think here the authors mean to challenge harmful/inequitable gender norms. On lines 100-101, the authors state: “There is a belief that the social fathering role represents a real opportunity to promote more positive gender identity.” Again, I do not think this is the right term here. Gender identity refers to the personal sense of one’s own gender (whether they identify as male, female or other). I do not think the authors intended to refer to this, and I’m not sure what a “positive gender identity” is. Instead of gender identify, did the authors mean to say equitable gender norms?

- I also think it would be useful for the authors to unpack the idea of social fathering more. How does it promote positive gender identity- again, do you mean equitable gender norms? I imagine as you mentioned above, the impact of social fathering on boys’ endorsement of equitable gender norms might not necessarily be positive- it depends on the social father’s views.

- Context of study: A little more information about what the iALARM task team does would be useful here and how the soccer clubs fit into this parent study. You say you analyzed iALARM meeting minutes but it remains unclear how are those relevant to your research questions. Who is included in these meetings? What is discussed?

Methods:

- As noted earlier, I think a rationale needs to be provided for why you didn’t also explore the perspectives of the boys themselves.

- Again, a rationale for reviewing the iALARM meeting minutes would be helpful. Why are these relevant to your research question? What information were you trying to learn from these meetings?

- Where were the interviews conducted?

Results:

- As noted previously, the authors need to provide more evidence of the gendered socialization within the soccer clubs.

- The quotes shared for the section on complexities around the social fathering role do not clearly illustrate how playing this role is “not easy” or that “there are complexities associated with playing it.” I suggest you add more quotes to better articulate what you mean by this.

- More quotes are needed to illustrate coaches frustrations with navigating tenuous dynamics with parents.

Reviewer #2: General comments

The manuscript is well written and scientific sound. It can be considered for publication in PloseOne journal.

I have the following few comments

Why did the authors conduct 11 interviews?

How did the authors recruit the participants for actual fieldwork?

Did the participants that attend interview informed about the research??/

The authors must correct error on line 422, they wrote “a” instead of “are”

Another error, “instead of couching” authors wrote” couches “

May I request the authors to account on how they applied reflexivity to enhance the quality of the results?

Results

The first theme and sub theme are specifically about coaches’ perceptions of adolescent boys’ problems / social problems boys are facing.

- Is TMC a social problem? Or behavioural aspects of adolescents who attend this rite of passage to manhood present social problems?

- The authors seem to think all the boys that attended TMC only start dagga, alcohol etc when they were in initiation school, however research shows that adolescent start these as early as 14 years or even younger.

- What are the ages of the adolescents that are being coached. TMC in Eastern Cape and Cape Town provinces is only allowed in boys from 18 years and older -if the coaches can prevent such behaviours until initiation; I would also recommend soccer avenues. line 497, some coaches said they are working with boys/men from 8-17 years this is before they go to initiation, do they also have boy who went before 18 years?

- This theme/sub theme doesn’t clearly articulate or describe the research questions from line 117-120

- Maybe the authors may consider defining what are social problems in their study first, before they attempt this theme/ subtheme

Line 485: Complexities around the social fathering role

LINE 497- 500 and 507-514: The excerpts presented are describing complexities of being a soccer coach. I am saying this because authors interviewed coaches. Coaches are biological fathers first; biological fathers are expected to support their boys. Now when these biological fathers become couches they are expected to fill in all these responsibilities including social fathering.

Reviewer #3: This is an important paper which seeks to assess how gender socialization of adolescent boys works in soccer

clubs, and whether there are opportunities for having gender transformative elements in that socialization.

The authors need to deal with few things to get the paper to a more solid place.

1. The paper uses concepts such as "gender socialization" of adolescent boys and "gender transformative" without clearly defining what they mean. Gender Socialization is the process of educating and instructing males and females as to the norms, behaviors, values, and beliefs of group membership as men or women. Is this what the coaches were doing? What ae the acceptable ways of being a boy in the study area-- to be fearless, competitive, heterosexual etc? I am curious to know what gender transformative socialization means in this context? It will be key to clarify what definitions of these concepts that the authors now apply to the coaches' work. Of course, soccer can be a platform for gender socialization, but is a also a very competitive sports that builds out tendencies for striving, endurance, not showing pain, violence and extreme hardiness in boys etc. Were these part of the virtues the coaches sought to instill in the boys? The focus of the coaches' engagement with the boys were on sexual and reproductive health, alcohol and drugs and involvement in criminal gangs. And in some cases supporting the boys to think about their life journeys and change their risky behavior and practices. But gender socialization also goes well beyond these to include traits and practices that are also harmful to boys.

2. The authors need to build a paragraph or two on the context of gender, particularly in relation to adolescence in SA or in the area of the study. The literature reviews lays out the debates and research on soccer and gender, but it is silent on the issue of gender socialization and gender and adolescence in the study area. Addressing this gap will add clarity to why this study is important.

3. The paper talks about the coaches' beliefs that traditional male circumcision (TMC) as a key mechanisms for socializing boys longer performs its intended socializing role. But what is it that TMC did in the past that it is was able to "properly" socialize boys in the past?

4: I think a more fundamental issue is the lack of a critical appraisal of the kind of gender socialization that these boys were receiving, that whether on one hand they were re-inscribing traditional notions of masculinity/gender that could also harm the boys, which soccer is globally known for.

5: What was the ultimate boy-product that these coaches wanted to see. This needs to come out clearly in the study. What additional social(ization)work is needed with these boys from the perspective of the coaches? This is not clear too.

6. PLOS authors have the option to publish the peer review history of their article (what does this mean?). If published, this will include your full peer review and any attached files.

Reviewer #1: No

Reviewer #2: No

Reviewer #3: No

---

## [Author Response · Author response to Decision Letter 0]

6 Jul 2022

The point-to-point response to reviewers' comments document is attached.

---

## [Decision Letter · Decision Letter 1]

11 Aug 2022

PONE-D-21-26547R1The potential of soccer clubs as avenues for gender transformative socialization of adolescent boys in Cape Town and Mthatha, South Africa: findings from a qualitative studyPLOS ONE

Dear Dr. Sikweyiya,

Please address reviewers' comments adequately. See for example, reviewer #3 who feels their comments have been parried. Please submit your revised manuscript by Sep 25 2022 11:59PM. If you will need more time than this to complete your revisions, please reply to this message or contact the journal office at plosone@plos.org. Please include the following items when submitting your revised manuscript:A rebuttal letter that responds to each point raised by the academic editor and reviewer(s). You should upload this letter as a separate file labeled 'Response to Reviewers'.A marked-up copy of your manuscript that highlights changes made to the original version. You should upload this as a separate file labeled 'Revised Manuscript with Track Changes'.An unmarked version of your revised paper without tracked changes. You should upload this as a separate file labeled 'Manuscript'.

We look forward to receiving your revised manuscript.

Kind regards,

Webster Mavhu

Academic Editor

PLOS ONE

Additional Editor Comments (if provided):

See comment from reviewer #3.

I am reviewer 3 and I feel my earlier comments were parried rather than addressed in any solid way. Authors should make an effort to adequately. address reviewers' comments.

Reviewers' comments:

Reviewer's Responses to Questions

**Comments to the Author**

1. If the authors have adequately addressed your comments raised in a previous round of review and you feel that this manuscript is now acceptable for publication, you may indicate that here to bypass the “Comments to the Author” section, enter your conflict of interest statement in the “Confidential to Editor” section, and submit your "Accept" recommendation.

Reviewer #3: (No Response)

2. Is the manuscript technically sound, and do the data support the conclusions?

Reviewer #3: Yes

3. Has the statistical analysis been performed appropriately and rigorously? 

Reviewer #3: N/A

4. Have the authors made all data underlying the findings in their manuscript fully available?

Reviewer #3: Yes

5. Is the manuscript presented in an intelligible fashion and written in standard English?

Reviewer #3: Yes

6. Review Comments to the Author

Reviewer #3: The authors have done their best to respond to the comments. But they could do better through critical engagement with their data and my earlier comments. I am reviewer 3 and I feel my earlier comments were parried rather than addressed in any solid way.

What rings very clear to me appears to be that the coaches are interested in supporting and molding out a soccer masculinity that will make their coaching careers successful. It may or may not be any conscious effort to mold a new gender equitable male youth in South Africa.

In my previous review, I asked : What is the ultimate boy-product that these coaches wanted to see? Hard as this question appears, it lies at the heart of transformative gender work and partly defines your work contribution to the field. It's difficult to appreciate the work of the coaches, if the authors did not ask them to articulate a clear and shared vision of this new type of male youth their social fathering intends to produce? Or if the coaches themselves had no vision of what the new young man they want to create will look like in real life. Were they just hoping to produce good and committed soccer youth? Several successful sportsmen are also known for abusive behaviour towards children, women and other men! So what were they looking to create ultimately?

The authors still appear to conflate mentorship and social fathering. They appear to proceed from the argument that fathering and parenting are inadequate and poor, and that 's the problem of these young men. But the young men described in the coaches' narratives appear to be a troubled lot, dealing with the structural fallout of persistent and overwhelmingly violent apartheid system and colonial history. Societal values have moved forward and fast , creating a vast army of youth with little preparation and asset to tap into emerging opportunities or compete well in the new society. These are youth left behind in a country with the world's worst GINI coefficients. It is not poor fathering. This context is key to understand before we blame it on fathering and seek a solution that merely tackles symptoms rather than causes.

Still, it appears to me that the football platform appears to offer some form of useful entry point to draw young people away from some vices, but real re-socialization into a new norms of gender requires key structural changes that incorporate a focus beyond sports. Work is also needed at the household level where the youth return to, but where the coaches have little leeway to carry that work through. And, it is also really not their work and mission. Efforts and initiatives that seek to identify entry points for gender transformative work but focus little on the household and other critical settings that nest youth will have limited impacts. There is still little engagement with the concept of transformative gender work. The definition used for this is taken for granted and not problematized. The authors' response to the question of how TMC socialized boys is mere romanticism. It assumes a perfect gender past! The study also needs to decolonize itself. It falls into the traps of assuming that gender values are an ideal.

7. PLOS authors have the option to publish the peer review history of their article (what does this mean?). If published, this will include your full peer review and any attached files.

Reviewer #3: No

---

## [Author Response · Author response to Decision Letter 1]

25 Sep 2022

Dear Editor, 

I have attached the Response to Reviewers document. 

Best wishes

Yandisa

---

## [Decision Letter · Decision Letter 2]

9 Nov 2022

PONE-D-21-26547R2The potential of soccer clubs as avenues for gender transformative socialization of adolescent boys in Cape Town and Mthatha, South Africa: findings from a qualitative studyPLOS ONE

Dear Dr. Sikweyiya,

Thank you for submitting your manuscript to PLOS ONE. After careful consideration, we feel that it has merit but does not fully meet PLOS ONE’s publication criteria as it currently stands. Therefore, we invite you to submit a revised version of the manuscript that addresses the points raised during the review process.  We have a few suggestions - see attached.Also, see if you can reduce word count from >10,400 words to 4-5,000. Please submit your revised manuscript by Dec 24 2022 11:59PM. If you will need more time than this to complete your revisions, please reply to this message or contact the journal office at plosone@plos.org. Please include the following items when submitting your revised manuscript:A rebuttal letter that responds to each point raised by the academic editor and reviewer(s). You should upload this letter as a separate file labeled 'Response to Reviewers'.A marked-up copy of your manuscript that highlights changes made to the original version. You should upload this as a separate file labeled 'Revised Manuscript with Track Changes'.An unmarked version of your revised paper without tracked changes. You should upload this as a separate file labeled 'Manuscript'.If applicable, we recommend that you deposit your laboratory protocols in protocols.io to enhance the reproducibility of your results. Protocols.io assigns your protocol its own identifier (DOI) so that it can be cited independently in the future. For instructions see: https://journals.plos.org/plosone/s/submission-guidelines#loc-laboratory-protocols. Additionally, PLOS ONE offers an option for publishing peer-reviewed Lab Protocol articles, which describe protocols hosted on protocols.io. Read more information on sharing protocols at https://plos.org/protocols?utm_medium=editorial-email&utm_source=authorletters&utm_campaign=protocols.

We look forward to receiving your revised manuscript.

Kind regards,

Webster Mavhu

Academic Editor

PLOS ONE

Journal Requirements:

Additional Editor Comments:

The manuscript is too long (over 10,400 words). We recommend articles do not exceed 4,000 words. This is flexible but exceeding this will impact upon the paper's 'readability'. I would recommend reducing the manuscript to at least 5,000 words. One way would be to truncate or reduce number of quotes. Ideally, a quote should be strengthening an idea.

Reviewers' comments:

Reviewer's Responses to Questions

**Comments to the Author**

1. If the authors have adequately addressed your comments raised in a previous round of review and you feel that this manuscript is now acceptable for publication, you may indicate that here to bypass the “Comments to the Author” section, enter your conflict of interest statement in the “Confidential to Editor” section, and submit your "Accept" recommendation.

Reviewer #3: (No Response)

2. Is the manuscript technically sound, and do the data support the conclusions?

Reviewer #3: Partly

3. Has the statistical analysis been performed appropriately and rigorously? 

Reviewer #3: N/A

4. Have the authors made all data underlying the findings in their manuscript fully available?

Reviewer #3: Yes

5. Is the manuscript presented in an intelligible fashion and written in standard English?

Reviewer #3: Yes

6. Review Comments to the Author

Reviewer #3: The paper can be published. I have done my best to push the authors to reflect deeply on what they are saying and not saying. The authors may wish to add robust and more elaborate caveats that show the deep limitations of their paper.

7. PLOS authors have the option to publish the peer review history of their article (what does this mean?). If published, this will include your full peer review and any attached files.

Reviewer #3: No

---

## [Author Response · Author response to Decision Letter 2]

23 Dec 2022

Dear Editor, 

The Response to Reviewers' comments has been uploaded as a separate file. 

Best wishes

Yandisa

---

## [Decision Letter · Decision Letter 3]

12 Jan 2023

Soccer clubs as avenues for gender transformative socialization of adolescent boys in Cape Town and Mthatha, South Africa: a qualitative study

PONE-D-21-26547R3

Dear Dr. Yandisa,

We’re pleased to inform you that your manuscript has been judged scientifically suitable for publication and will be formally accepted for publication once it meets all outstanding technical requirements.

Kind regards,

Nelsensius Klau Fauk, S.Fil., M., MHID, MSc, PhD

Academic Editor

PLOS ONE

Additional Editor Comments (optional):

Reviewers' comments:

Reviewer's Responses to Questions

**Comments to the Author**

1. If the authors have adequately addressed your comments raised in a previous round of review and you feel that this manuscript is now acceptable for publication, you may indicate that here to bypass the “Comments to the Author” section, enter your conflict of interest statement in the “Confidential to Editor” section, and submit your "Accept" recommendation.

Reviewer #3: All comments have been addressed

2. Is the manuscript technically sound, and do the data support the conclusions?

Reviewer #3: Partly

3. Has the statistical analysis been performed appropriately and rigorously? 

Reviewer #3: N/A

4. Have the authors made all data underlying the findings in their manuscript fully available?

Reviewer #3: Yes

5. Is the manuscript presented in an intelligible fashion and written in standard English?

Reviewer #3: Yes

6. Review Comments to the Author

Reviewer #3: No further comment. The paper can be published. I was expecting a more thorough-going discussion of the limitations. The findings would have benefited from a more robust contextualization within South African social processes. Just more critical engagement with what the data is saying or not saying would have been great.

7. PLOS authors have the option to publish the peer review history of their article (what does this mean?). If published, this will include your full peer review and any attached files.

Reviewer #3: No

---

## [Editor Report · Acceptance letter]

23 Jan 2023

PONE-D-21-26547R3 

Soccer clubs as avenues for gender transformative socialization of adolescent boys in Cape Town and Mthatha, South Africa: a qualitative study 

Dear Dr. Sikweyiya:

I'm pleased to inform you that your manuscript has been deemed suitable for publication in PLOS ONE. Congratulations! Your manuscript is now with our production department. 

Kind regards, 

on behalf of

Dr. Nelsensius Klau Fauk 

Academic Editor

PLOS ONE